# Optimal Water Level Management for Mitigating GHG Emissions through Water-Conserving Irrigation in An Giang Province, Vietnam

**DOI:** 10.3390/s22218418

**Published:** 2022-11-02

**Authors:** Satoshi Ogawa, Kyosuke Yamamoto, Kenichi Uno, Nguyen Cong Thuan, Takashi Togami, Soji Shindo

**Affiliations:** 1Japan International Research Center for Agricultural Sciences (JIRCAS), Crop, Livestock and Environment Division, 1-1 Ohwashi, Tsukuba 305-8686, Ibaraki, Japan; 2Nosho Navi, Inc., 339-3, Satsuma-cho, Hikone-shi 521-1147, Shiga, Japan; 3e-kakashi Section, CPS Technology Planning Department, Technology Planning & Development Division, Service Planning Technology Division, SoftBank Co., 1-7-1 Kaigan, Minato-ku, Tokyo 105-7529, Tokyo, Japan; 4Rural Development Division, JIRCAS, Tsukuba 305-8686, Ibaraki, Japan; 5College of Environment and Natural Resources, Can Tho University, Can Tho 94000, Vietnam

**Keywords:** alternate wetting and drying, multiple drainage, water level, rice yield, climate change

## Abstract

Rational water and fertilizer management approaches and technologies could improve water use efficiency and fertilizer use efficiency in paddy rice cultivation. A promising water-conserving technology for paddy rice farming is the alternate wetting and drying irrigation system, established by the International Rice Research Institute. However, the strategy has still not been widely adopted, because water level measurement is challenging work and sometimes leads to a decrease in the rice yield. For the easy implementation of alternate wetting and drying among farmers, we analyzed a dataset obtained from a farmer’s water management study carried out over a three-year period with three cropping seasons at six locations (*n* = 82) in An Giang Province, Southern Vietnam. We observed a significant relationship between specific water level management and the rice yield and greenhouse gas emissions during different growth periods. The average water level during the crop period was an important factor in increasing the rice yield and reducing greenhouse gas emissions. The average water level at 2 days after nitrogen fertilization also showed a potential to increase the rice yield. The greenhouse gas emissions were reduced when the number of days of non-flooded soil use was increased by 1 day during the crop period. The results offer insights demonstrating that farmers’ implementation of multiple drainage during whole crop period and nitrogen fertilization period has the potential to contribute to both the rice yield increase and reduction in greenhouse gas emissions from rice cultivation.

## 1. Introduction

Rice (*Oryza sativa* L.) is an important staple food which is cultivated in many countries in Asia, Africa, Latin America, and the Caribbean, and it supplies 35–60% of the dietary calories consumed by almost half of the world population [1]. In Vietnam, one of the major rice producers globally, the annual rice cultivation area and production amount in 2019 were 7.5 million ha and 44 million t, respectively [2]. The Mekong Delta region in southwestern Vietnam has favorable water and climatic conditions that enable farmers to cultivate rice three times a year and accounted for 55.2% of the rice production in 2017 [2]. However, rice growth and development require substantial amounts of freshwater and generate huge amounts of methane (CH_4_) emissions during the cultivation period. Both are major problems in Vietnam and are exacerbated climate change and increased water consumption in populated area [3,4].

CH_4_ emissions from rice cultivation also represent a serious environmental issue in Vietnam. Almost 50% of the national agricultural greenhouse gas (GHG) emissions were generated by rice cultivation, representing approximately 16% of the national anthropogenic GHG emissions, including land use change and forestry [5]. In addition, the total rice production has been increasing steadily in recent decades following the introduction of electric pump irrigation, along with new rice varieties and fertilizers. However, farmers’ incomes have not matched the increase in production because of the poor grain quality and high production costs associated with the introduction of new irrigation systems and chemical fertilizers [6]. Consequently, researchers are developing novel water management systems to improve the rice yield and grain quality and to facilitate sustainable rice production by improving the water and fertilizer use efficiency [7].

Alternative wetting and drying (AWD) is a water-conserving irrigation system that was established by the International Rice Research Institute (IRRI) in the Philippines. The adoption of this practice has been recommended for irrigated rice cultivation areas with water shortages in South and Southeast Asia [8]. In the central plain of Thailand, it is also reported that AWD increased the rice yield by 7% in the dry seasons and by 15% in the wet seasons when compared with traditional continuous flooding (CF) irrigation [9]. Therefore, AWD could be disseminated elsewhere if the associated rice yield increase were to be demonstrated in various countries. This management approach has spread extensively in An Giang Province, Vietnam, owing to the establishment of full-dike systems across the province [10]. One potential reason for the high adoption rate of AWD is that it may enhance the rice yield compared to the yields obtained under the conventional irrigation system (i.e., CF) [11]. For example, the practice of AWD water management in a rice cultivation field in An Giang Province increased the rice yield by 2–5% compared to the traditional CF area, in addition to reducing CH_4_ emissions by 21–74% [12]. AWD management controls irrigation according to the water level. However, in real farmers’ paddy fields, water management is performed depending on the convenience and skill of the farmers. Sometimes, water management is performed through mid-season drainage and/or multiple drainage (MD), in which the water level naturally declines below the soil surface. These simplified water management systems also resulted in an increase in the rice yield and reduction in the GHG emissions in Mekong Delta [13], China [14], and Vietnam [15]. However, there are a few successful cases of the adoption of the AWD or MD practices in rice cultivation. It was also reported that this had no effect or negative effects on the rice yield [16]. This is the reason why there are significant variations in the farmers’ skills and other field management practices, with respect to factors such as the nitrogen content of the soil, wood management, and chemical applications, as observed when all the field management practices were conducted by the farmers themselves [17]. Therefore, it important to analyze and identify specific practices, such as water and fertilizer management techniques, through farm data. There is little research on farmers’ simultaneous practices, and it remains unclear whether they could enhance their rice yields considerably. This kind of approach could contribute to the acceleration of AWD management in other rice cultivation areas through a reduction in the risks involved in water management and create new values with regard to GHG emissions. In the future, the combination of this technique with new technology, such as the internet of things (IoT), for automatic water level management may minimize farmers’ efforts while saving time and increasing both the rice yield and profit. However, at this moment, there is not sufficient scientific evidence regarding the relationship between specific water level management and the rice yield under farming conditions.

In addition to AWD water management, shallow water management is recommended for tropical regions in Vietnam and other countries [18]. In particular, the water level at the time of nitrogen fertilizer application is a critical factor to consider for the improvement of rice productivity [19] and reduction in GHG emissions [20]. According to the “One Must Do, Five Reductions (1M5R)” program, certified by the Vietnam Ministry of Agriculture and Rural Development in 2013, before applying nitrogen fertilizer as a top dressing, the water level should be maintained at 3–5 cm above the soil surface during the first and second nitrogen fertilizer applications, and 1–3 cm above the soil surface during the third nitrogen fertilizer application. The application of nitrogen fertilizer to shallow-flooded soil facilitates the mixing of the fertilizer with the soil and prevents fertilizer losses. If it is not possible to drain the field before the nitrogen fertilizer application, the fertilizer can be dispersed in standing water, followed by an interculture operation. In such a case, the water should not be allowed to leave the field for at least 24 h. Still, in China, there was no significant effect of the interaction between the water and nitrogen on the rice yields and nitrogen use efficiency [21], because the farmers’ practice of both water and nitrogen application management is varied, depending on their skill and experience. Therefore, it is necessary to explore appropriate nitrogen application strategies and rates through water-conserving irrigation based on specific water level conditions at the farm site. Moreover, the ways in which the control and optimization of farmers’ water level management at different nitrogen fertilizer application rates influences the rice yield are yet to be addressed in the case of water-conserving irrigation systems such as AWD and MD. The objective of the present study was to identify specific water level management practices associated rice yield improvements and reductions in GHG emissions under farming conditions in An Giang Province, Vietnam, over the course of nine consecutive rice growing seasons. We generated the features of specific water levels to simplify the raw water level data in the whole cultivation period, and we found that they have significant correlations with the rice yields and GHG emissions. Determining the critical period for specific water level management using simple feature values could facilitate rice yield improvements and the mitigation of GHG emissions in the case of farmers’ conventional agronomic practices.

The results of the present study can offer effective approaches to the introduction of science-based agriculture, which could enhance the rice yield by improving the water level management of paddy rice cultivation systems in South Vietnam and in other regions with similar environmental conditions. In the future, the determination of specific water level management techniques may be helpful for the introduction of the automation of water management, which can contribute to both rice yield improvements and the mitigation of GHG emissions in the context of water-conserving irrigation in An Giang Province, Vietnam.

## 2. Materials and Methods

### 2.1. Data Source

Agronomic data were used and collected from field experiments that were carried out at six locations in the full-dike areas (Chau Thanh, Cho Moi, Thoai Son, Tri Ton, Chau Phu, and Tinh Bien) across An Giang Province, Vietnam (Figure 1). A total of 36 experimental field data from Chau Thanh, Cho Moi, and Thoai Son were as the same as those used by Uno et al. (2021) [15]. An additional 46 experimental field data were collected to cover the whole An Giang Province with same method as that used by Uno et al. (2021) [15]. Six locations were selected according to their different soil types and the skills of the farmers, who can manage two water management schemes, including CF and MD, allocated to two neighboring fields managed by the same farmer at each location. Agronomic data were collected over three rice growing seasons: the spring–summer season (SS, April–August), autumn–winter season (AW, August–December), and winter–spring season (WS, winter–spring, December–April) from April 2015 to April 2018. The monthly temperature and precipitation in An Giang Province during the whole experimental period are shown in Figure 2. In the target region, there is usually a 4–5-month dry season from December to April, sometimes including May, and a 6–7-month rainy season from May to November (Figure 2). A direct sowing method was used, with the selected rice variety based on the recommendations of agricultural extension centers for each cropping season (Table 1). Fertilizer was applied based on the rice variety and soil fertility of each location, but the applied amounts were similar between the two water management practices (Table 1). All the other agronomic practices, such as the soil preparation, including the management of the rice straw and pest and disease control, were also consistent with the conventional local practices of farmers and conducted with the help of extension center staff. Consequently, the applied fertilizer rates and straw management approaches varied across seasons and locations. The locations, soil types, rainfall, and field management practices are listed in Table 1. In total, 82 field data were collected, including both the CF and MD water management schemes. Every morning, the farmers measured the water level using a water gauge formed of a polyvinyl chloride pipe that was installed near the GHG measuring area in each rice cultivation field. The gauges were installed to measure the water level 15 cm below the soil surface. Following to the method described by Taminato and Matsubara (2016) [12], GHG measuring was conducted every week in three replications over a 1 m × 1 m area in each rice cultivation field. The harvest was conducted in the same area as the GHG measuring point, avoiding poor or over-luxuriant growth. After wind drying, the paddy rice yield was calculated through the calibration of the moisture content. The average values of the water level, rice yield, and GHG emissions such as CH_4_ and nitrous oxide (N_2_O) were used for further analyses to investigate the relationships between them.

The average monthly temperatures are shown as orange lines, and the average monthly maximum and minimum temperatures are shown as broken lines, with red and blue lines, respectively. The volume of precipitation is shown in the bar chart.

### 2.2. Feature Selection of Specific Water Levels and Statistical Analysis

The effects of the feature selection of specific water levels on the rice yield and GHG (CH_4_ and N_2_O) emissions were evaluated. To investigate the relationships between them, we first generated feature values from the water level data. A Pearson’s correlation coefficient analysis was conducted to investigations correlations between the determinate feature values and the rice yield or GHG emissions (Table 2). Linear regression was also conducted to assess the quantitative relationships. Logarithmic conversion was applied to the CH_4_ emission values before the linear regression. The statistical analyses were conducted using Python v3.8.10 “https://www.python.org/downloads/release/python-3810/ (accessed on 28 September 2022)” and the SciPy library v1.6.2 “https://www.scipy.org/ (accessed on 28 September 2022)”. An analysis of variance (ANOVA) was performed using R v3.6.3 “https://cran.r-project.org/bin/macosx/ (accessed on 28 September 2022)” to assess the effects of the season, soil type, location, and their interactions on the rice yield and CH_4_ emissions.

## 3. Results

### 3.1. Spatial and Temporal Variation in the Rice Cultivation

The cultivation conditions in the present study varied in terms of the soil type, season, rainfall, rice variety, and fertilizer and water management (Table 1). Inorganic fertilizers, such as N, phosphorus, and potassium were sometimes applied at more than double the usual rate in different locations and seasons because of the soil fertility conditions (Table 1). However, there was no significant correlation between the rice yield and GHG emissions, excluding under specific water management conditions (Table 2). The accumulated temperature was correlated with the crop period and duration of the final drainage. The frequency of the pump operation was correlated with the number of days of negative water levels and the average water level when nitrogen fertilizer was applied. Only three water level management conditions were correlated with the rice yield (Table 2).

Significant differences (ANOVA) in the rice yield and CH_4_ emissions in the An Giang Province over the three rice cultivation seasons are shown in Table 3. The season, water management, and soil type had major effects on the rice yield (*p* < 0.05). The water management had the main effects on the CH_4_ emissions, and the season and location had significant interactive effects on the CH_4_ emissions (*p* < 0.01). However, there was no significant difference in the N_2_O emissions between the CF and MD management practices, because the nitrogen fertilizer was often applied under flooded conditions. Similar trends were also reported by [13,15]. Therefore, only the CH_4_ emissions were used to analyze the relationship between the water managements and GHG emissions. In response to seasonal variations in the water levels, most CH_4_ emissions were lower in the case of MD management than in CF management. Significant differences were observed in the rice yield and CH_4_ emissions between the MD and CF management practices, with *p*-values of 0.0039 and 0.0019, respectively (Table 3).

### 3.2. Water Management

Most of the observed water levels for the CF management practice were maintained under flooded conditions (i.e., >0 cm above the soil surface) regardless of the season, excluding fields 8, 60, 68, 70, and 72. The water levels for the MD management varied because of the soil type and seasonal conditions. However, the water level was lower than 0 cm below the soil surface at least once before water drainage was applied for the harvest. Only the water levels in fields 57 and 79 were maintained as flooded because of heavy rainfall and poor drainage.

### 3.3. Relationships between Specific Water Management Practices and the Rice Yield

Four specific water levels—the average water level during the crop period and during a certain period before and after nitrogen fertilization, the number of days when the water level was less than 0 cm, and the accumulated negative water level during the crop period—were determined by the feature selection. To determine the specific influence of the water management approach on the rice yield, a Pearson’s correlation coefficient analysis was conducted (Table 2). Following the result of the analysis, we determined that specific water management approaches included the average water level during crop period and number of days when the water level was less than 0 cm, and the accumulated negative water level exhibited a significant correlation with the rice yield (Figure 3; R = −0.335, 0.388, and −0.390, respectively, with *p*-values < 0.01). We focused on the water level during the nitrogen application period, including the two days before and after. We measured that the average water levels before and two days after the nitrogen fertilizer application to determine the date on which the water level was highly correlated with the rice yield (Figure 4).

The average water levels at one and two days after the nitrogen fertilizer application exhibited the highest correlations with the rice yield (Figure 4; R = −0.437, *p* < 0.001). The results indicated that the average water level was the most important management factor, and the rice yield could increase by 170 kg per ha when the water level was reduced by 1 cm 1–2 days after N fertilization (Figure 4f). The average water level during the cropping period, from sowing to the final drainage for the harvest, was also a major factor influencing the rice yield and could increase it by up to 120 kg per ha when the water level was decreased by 1 cm (Figure 3a). Both the number of days when water level was less than 0 (Figure 3b) and the accumulated water level during the crop period (Figure 3c) also exhibited significant correlations with the rice yield (*p* < 0.01); however, their coefficients (0.04 and −0.00, respectively) were very small.

### 3.4. Relationships between Specific Water Management Practices and CH_4_ Emissions

Specific water management practices were observed to reduce the CH_4_ emissions in different crop seasons (Figure 5 and Figure 6). The average water level during the cropping period, number of days when the water level was less than 0 cm, and accumulated negative water level exhibited significant correlations with the CH_4_ emissions (R = 0.433, −0.433, and 0.324, respectively, with *p*-values < 0.01). In addition, the average water level during the cropping period (Figure 5a) exhibited a significant correlation with the CH_4_ emissions. Tt could reduce the CH_4_ emissions by 9% when the average water level was 1 cm lower during the whole crop season. The number of days of non-flooded soil (Figure 5b) also exhibited a significant correlation with the CH_4_ emissions, which could potentially be reduced by 3% when it was increased by 1 d during the whole cropping period, from sowing to the final water drainage. Furthermore, there was a correlation with the accumulated negative water levels during the cropping period, although it was smaller than the above-mentioned water levels (Figure 5). The average water levels before and two days after the nitrogen fertilizer application were also analyzed. Only the average water level before the nitrogen fertilizer application showed the highest correlation with the CH_4_ emissions (Figure 6; R = 0.268–0.276 with *p*-values < 0.05), which implies that the water level management at 1–2 days before nitrogen fertilization could potentially reduce GHG emissions by 4–6% when the water level was 1 cm lower (Figure 6a–c).

## 4. Discussion

This study investigated the effects of specific water level management in enhancing rice yield improvements and the reduction in GHG emissions under farming conditions across six locations in An Giang Province, Vietnam, and elucidated the relationships between them. We identified specific water level management practices for increasing the rice yield and decreasing CH_4_ emissions. Since excess water above the optimal level could also affect the rice yield and mitigation of GHG emissions, provisions for draining the excess water are just as important as irrigation. However, to obtain the correct water quantity and timing, farmers must collect the water level measurements more frequently and irrigate as needed. Farmers should also have autonomous access to irrigation with a sufficient water supply to be able to pump water as per the field requirements, which is a difficult job for farmers. Therefore, it is an obstacle for them. To solve these problems, we aimed to apply IoT technology to address some of the challenges faced by the farmers regarding the optimal water level management [22]. Water level monitoring by cloud-based management can help farmers to understand the actual and recommended water levels and determine the optimal time to irrigate the rice, as needed, through mobile phone applications. This preliminary study examined the effects of specific water level management techniques on the rice yield and GHG emissions under farming conditions in An Giang Province, Vietnam. We are currently preparing a new experiment to validate the results of this study, using a new scaling factor with IoT to improve the rice yield and reduce the GHG emissions from paddy fields by using cloud-based management through mobile phone applications. This kind of approach could reduce the risks for farmers in pursuing IoT water level management technology while providing substantial benefits, such as reduced GHG emissions and the sustainable use of water. Additionally, it would also contribute to the creation of new values for carbon credits through water level monitoring as traceability information.

### 4.1. Water Level Management

The 1M5R manual provides a detailed water management procedure for use in Vietnam during the fertilizer application period [23]. Agricultural extension centers in An Giang Province recommend a pumped-up water level of 1–3 cm before the application of nitrogen fertilizer [23]. In a previous study, shallow water levels provided better conditions for rice growth, including the rice yield, while the effect of nitrogen fertilizer on the rice yield was not clear [18]. In China, there was no significant effect of the timing of nitrogen fertilization or its interaction with the water level on the rice yield [24].

In the present study, two water management schemes, CF and MD, were implemented at each location. However, the implementation of MD is often challenging due to poor drainage, especially in the rainy season. Notably, we observed specific water management approaches that could improve the rice yield and reduce CH_4_ emissions, even though there was no significant correlation of the fertilizer management practice with both the applied amount and timing. A similar trend was shown by reports on AWD’s effect on the rice yield and GHG emissions during a triple-cropping season in different farms of the Mekong Delta [13,25]. Such observations can facilitate voluntary action using relatively simple approaches to improve the rice yield and reduce CH_4_ emissions based on optimal water management. Further effects of water management during the nitrogen fertilization period on the rice performance, including the nitrogen use efficiency, should be examined in future.

### 4.2. Effect of Multiple Drainage on Rice Yield

In the present study, water management significantly affected the rice yield. There are several reports on significant increases in the rice yield under MD in the Mekong Delta [13,26]. Repeated drainage may induce favorable conditions for rice growth. Oxygen supply to the soil through MD management promotes nitrification, essential for the conversion of nitrogen forms from NH_4_^+^ to NO_3_^−^. A mixed supply of NH_4_^+^ and NO_3_^−^ resulted in positive impacts on the rice yield compared with the application of the sole nitrogen source [27]. Highly reductive soil conditions fostered by MD management are also helpful for avoiding lodging and climate resistance through the inhibition of the substances that are harmful to production, such as hydrogen sulfide and organic acids, in addition to enhancing root elongation [28]. However, according to the results of a previous meta-analysis [29], neither AWD nor MD increased the rice yield in China [30], including the Mekong Delta in Vietnam [31] and also Thailand [32], where this finding was especially noticeable because of water stress due to inappropriate AWD or MD implementation. Water stress is not only drought stress; it also includes deep-water issues related to the shortage of drainage systems in the paddy field. In fact, the effect on the rice yield increase of MD is higher in the wet season compared to the dry season [9]. It was reported that the direct sowing method can also mask the positive effects of MD on the rice yield due to the spatial variability in the rice growth [33]. To minimize this effect, this study set up 1 m × 1 m harvesting areas after the crop establishment. The results were obtained under these ideal crop establishment conditions, and the area was narrow so as to represent the normal yield of a field [15]. Therefore, further studies should employ both narrow and larger-scale yield-measuring areas to determine the quantitative effects of water level management under farming conditions. However, large-scale yields are also affected by heterogeneous environment factors, such as the water level, leveling, and soil fertility and type, requiring precise and frequent measurement during the rice cultivation period.

### 4.3. Effect of Multiple Drainage on GHG Emissions

The number of days when the water level was less than 0 cm from the soil surface had a negative correlation with the rice yield. However, compared to the previous results [12], the coefficient of determination observed in the present study was too low because of uncontrolled factors, such as farmers’ skills in adjusting the water depth threshold or irrigation and drainage timing. It was also reported that MD could decrease the CH_4_ emissions when compared with CF, as the control, in the same period in rice cultivation fields in the Mekong Delta, Vietnam [15,33]. However, it is difficult to determine the exact water level at which the growth period significantly affects CH_4_ emissions. Following a comparison between the CH_4_ emissions and the number of days when the water level was less than 0 cm and the accumulated negative water level during the crop period, it is important to increase the number of days of negative water levels (less than 0 cm) rather than the accumulated water level so as to reduce CH_4_ emissions. Furthermore, the average water level at the time of nitrogen fertilizer application was highly correlated with the improvement in the rice yield after the nitrogen application. Conversely, the water level before the nitrogen application was highly correlated with the reduction in CH_4_ emissions. Further studies are required to comprehensively clarify the relationships between these factors. Such findings could facilitate the management of water stress caused by inappropriate AWD or MD implementation and may offer appropriate indicators for water management activities in the context of paddy rice production, including the use of CH_4_ emission ratios of a treatment practice to a baseline practice in the GHG inventory systems developed by the Intergovernmental Panel on Climate Change (IPCC) [34] and targets for the mitigation of GHG emissions based on Vietnam’s Nationally Determined Contribution [35].

## 5. Conclusions

The present study examined the effects of specific water level management practices on the rice yield and GHG emissions under farming conditions at six locations in An Giang Province, Vietnam. We observed significant relationships between the factors and found that specific water management practices could increase the rice yield and reduce CH_4_ emissions. The control of the average water level during whole crop period has potential to both increase the rice yield and reduce CH_4_ emissions. Furthermore, the results can provide a basis for the dissemination of relatively simple, optimal water level management practices through water-conserving irrigation in Vietnam and regions with a similar climate and agronomic environment. However, to apply appropriate water levels at appropriate timings, farmers must regularly measure the water levels and irrigate as required. Farmers should also have independent access to irrigation with an adequate water supply to be able to pump water whenever their fields require it, which could be a challenge for farmers. In the future, we will conduct a study to validate the results of the present study, as novel indicators to be adopted in rice cultivation activities targeting the improvement of the rice yield and reduction in GHG emissions.

## Figures and Tables

**Figure 1 sensors-22-08418-f001:**
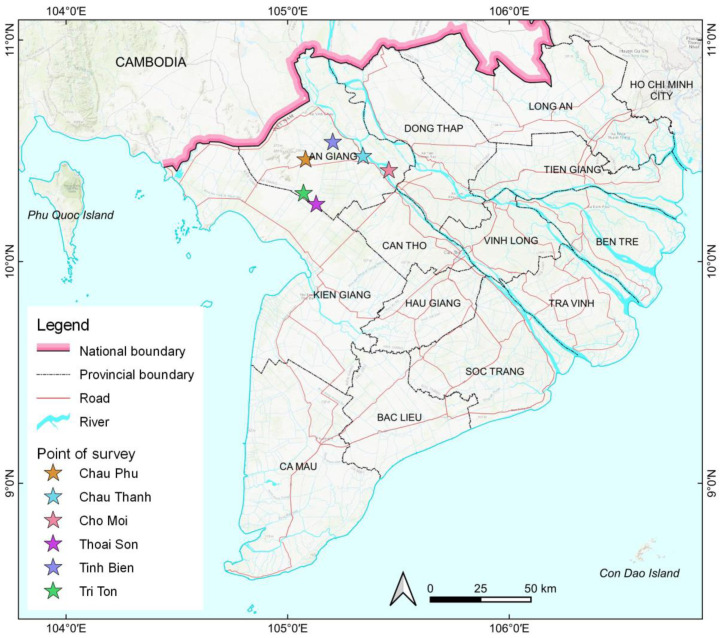
Locations of the study.

**Figure 2 sensors-22-08418-f002:**
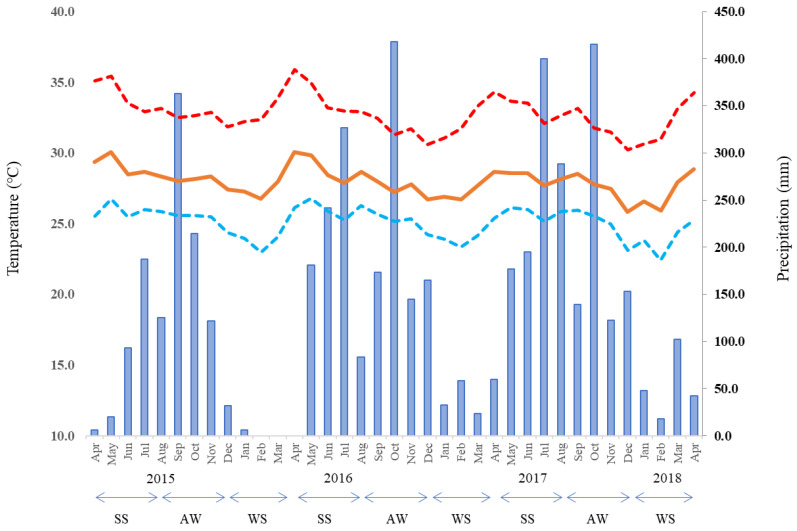
Monthly temperature and precipitation during the study period according to the representative meteorological stations in An Giang Province.

**Figure 3 sensors-22-08418-f003:**
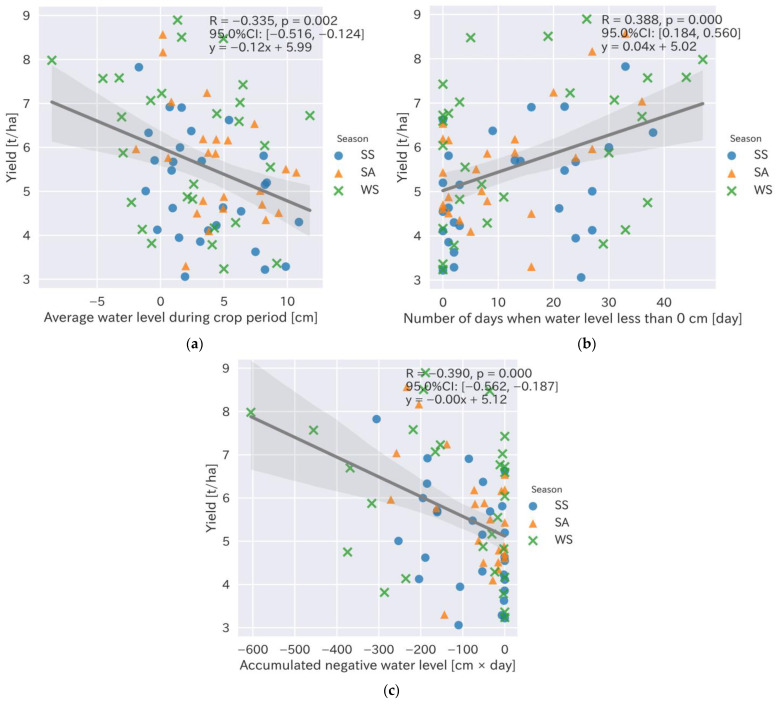
Relationship between the rice yield and water level during the crop cultivation period. (**a**) Average water level during the crop cultivation period (cm); (**b**) days of negative water levels during the crop cultivation period (days); (**c**) accumulated negative water level during the crop cultivation period (mm). Circles, triangles, and crosses show data from the spring–summer (SS), summer–autumn (SA), and winter–spring (WS) seasons, respectively.

**Figure 4 sensors-22-08418-f004:**
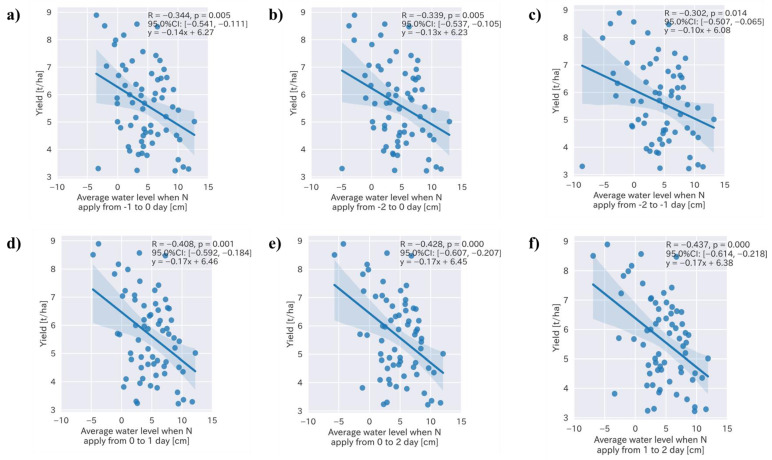
Relationships between the rice yield and average water level two days before and after nitrogen fertilizer application.

**Figure 5 sensors-22-08418-f005:**
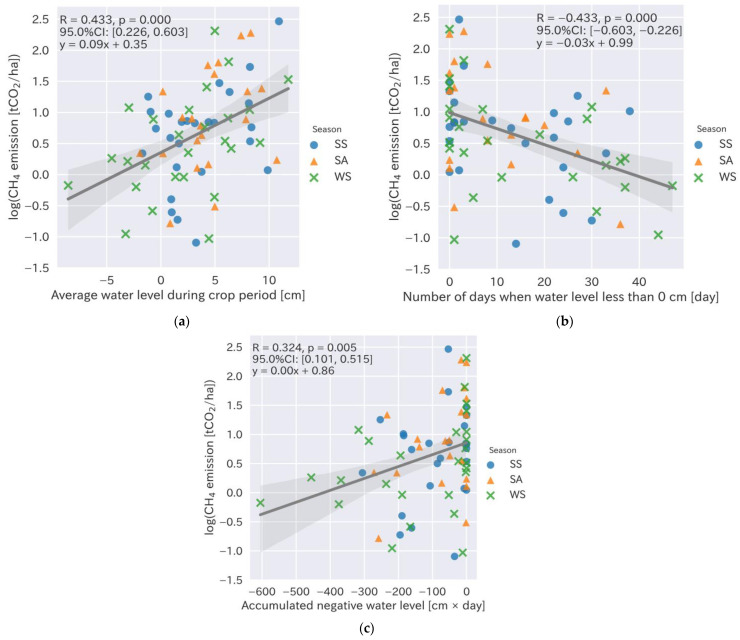
Relationship between methane emissions and specific water management practices during the cropping period and rice yield. (**a**) Average water level during the cropping period (cm); (**b**) days of negative water levels during the cropping period (days); (**c**) accumulated negative water level during the cropping period (mm). Circles, triangles, and crosses show data from the spring–summer (SS), summer–autumn (SA), and winter–spring (WS) seasons, respectively.

**Figure 6 sensors-22-08418-f006:**
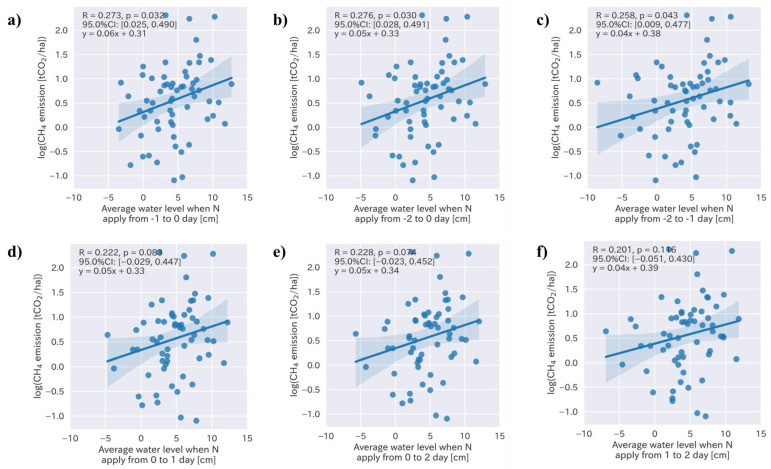
Relationship between methane emissions and average water levels two days before and after nitrogen fertilizer application.

**Table 1 sensors-22-08418-t001:** Location, soil type, rainfall, and field management of six experimental sites.

District		Chau Thanh	Cho Moi	Thoai Son	Tri Ton	Tinh Bien	Chau Phu
Latitude		10°28′27.7″ N	10°24′41.1″ N	10°15′29.1″ N	10°18′27.5″ N	10°32′20.0″ N	10°27′37.5″ N
Longitude		105°20′24.1″ E	105°27′25.9″ E	105°07′45.7″ E	105°04′19.6″ E	105°12′15.4″ E	105°04′53.4″ E
Soil type		Alluvial soil	Alluvial soil	Acidic soil	Acidic soil	Alluvial soil	Acidic soil
Rainfall (mm)							
2015	SS	299.5	190.1	452.1	277.0	n.d.	n.d.
	AW	711.8	588.4	718.3	n.d.	n.d.	n.d.
	WS	18.6	23.1	24.4	0.0	n.d.	n.d.
2016	SS	638.1	223.3	691.4	419.2	n.d.	n.d.
	AW	675.2	530.5	739.6	722.7	n.d.	n.d.
	WS	119.0	276.4	86.1	51.6	n.d.	n.d.
2017	SS	766.1	365.8	882.7	453.4	403.4	299.6
	AW	756.3	587.2	891.7	577.5	497.6	n.d.
	WS	108.3	67.8	166.2	125.0	143.2	165.3
Field management							
Variety							
2015	SS	OM6976	IR50404	IR50404	OM6976	n.d.	n.d.
	AW	OM5451		Jasmine	n.d.	n.d.	n.d.
	WS	OM4900		IR50404	IR50404	n.d.	n.d.
2016	SS	OM4900		IR50404	IR50404	n.d.	n.d.
	AW	OM5451		OM5451	IR50404	n.d.	n.d.
	WS	OM7347		IR50404	OM6976	n.d.	n.d.
2017	SS	Jasmine		Sticky rice	OM5451	OM9577	AGPPS114
	AW	OM4900		OM5451	OM5451	OM5451	n.d.
	WS	OM4900		IR50404	OM5451	OM5451	Jasmine 85
Inorganic fertilizer							
N (kg N ha^−1^)		96–264	14–37	92–159	36–95	5–13	83–117
P (kg P ha^−1^)		110–341	25–35	120–200	60–145	8	0–120
K (kg K ha^−1^)		30–133	5–30	0–153	29–130	8	80–150

SS: spring–summer season, April–August; AW: autumn–winter season, August–December, WS: winter–spring season, December–April.

**Table 2 sensors-22-08418-t002:** Correlation coefficient between the field managements, rice yield, and GHG emissions.

	Accumated Temperature	Precipitation	Total N	Number of N Applications	Average Water Level When N Applied	Crop Period	Final Drainage Period	Duration of Final Drainage	Days of Negative Water Levels	Number of Drainages	Accumulative Negative Water Level	Accumulative Water Level	Average Water Level	Frequency of Pump Operation	Cost of Pump Operation	CH_4_ Emissions	N_2_O Emissions
Precipitation	0.405																
Total N	0.125	−0.128															
Number of N applications	−0.103	−0.240	0.359														
Average water level when N applied	−0.154	0.035	−0.142	−0.040													
Crop period	**0.965**	0.306	0.149	−0.088	−0.169												
Final drainage period	0.363	0.188	0.338	−0.007	0.199												
Duration of final drainage	**0.693**	0.163	−0.113	−0.084	−0.326	−0.375											
Days of negative water levels	−0.105	−0.114	0.033	0.168	−0.566	−0.469	0.281										
Number of drainages	0.249	−0.086	−0.171	0.089	−0.368	−0.200	0.491	0.320	0.625								
Accumulative negative water level	0.186	0.206	0.031	−0.172	0.458	0.456	−0.204	−0.380	−0.904	−0.595							
Accumulative water level	0.103	0.142	0.127	−0.198	0.573	0.491	−0.262	−0.273	−0.720	−0.476	0.641						
Average water level	0.149	0.186	0.071	−0.225	0.570	0.508	−0.254	−0.357	**−0.892**	−0.583	**0.890**	**0.917**					
Frequency of pump operation	0.077	−0.159	−0.088	−0.280	**0.383**	0.189	−0.027	−0.237	−0.464	-0.181	0.342	0.314	0.373				
Cost of pump operation	−0.129	−0.159	0.250	−0.204	0.259	0.202	−0.244	0.031	−0.207	−0.103	0.032	0.186	0.134	0.587			
CH_4_ emissions	−0.046	0.159	0.063	−0.166	0.006	0.000	−0.067	−0.118	−0.165	−0.114	0.107	0.067	0.091	0.210	0.310		
N_2_O emissions	−0.019	−0.212	−0.068	0.060	0.139	0.021	0.111	−0.065	0.013	0.039	−0.111	−0.079	−0.097	0.131	0.122	0.054	
Rice Yield	0.112	−0.214	0.234	−0.123	**−0.341**	0.021	0.143	1.000	**0.410**	0.320	**−0.380**	−0.273	−0.357	−0.237	0.031	−0.118	0.128

Bold letters show significant differences at *p*-values < 0.05.

**Table 3 sensors-22-08418-t003:** Significant differences in the yield and methane emissions in the An Giang region over three rice cultivation seasons.

		Rice Yield					CH_4_ Emissions					N_2_O Emissions				
	Df	Sum Sq	Mean Sq	F Value	Pr (>F)		Sum Sq	Mean Sq	F Value	Pr (>F)		Sum Sq	Mean Sq	F Value	Pr (>F)	
Season	2	12.35	6.173	2.928	0.0608	ns	12.9	6.451	1.207	0.3058	ns	42.7	42.7	3.664	0.0605	ns
Water management	1	17.09	17.092	8.956	0.00385	**	61.7	61.7	11.61	0.00111	**	0.3	0.32	0.03	0.8633	ns
Soil type	1	6.66	6.663	3.16	0.0803	ns	21.8	21.794	4.079	0.0477	*	14	6.99	0.6	0.5523	ns
Water management × soil type	1	34	34	6.398	0.01375	**	0.13	0.126	0.066	0.79811	ns	3.9	3.94	0.367	0.5469	ns
Residuals	63	132.84	2.109				336.6	5.343				675.9	11.65			

* and ** indicate significance at the 0.05 and 0.01 levels, respectively; ns indicates no significance.

## Data Availability

Not applicable.

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
