# Peer review of "Optimal Water Level Management for Mitigating GHG Emissions through Water-Conserving Irrigation in An Giang Province, Vietnam"

_sensors, 2022, doi:10.3390/s22218418_

Round 1

Reviewer 1 Report

The authors investigated the relationship between water levels of paddy fields and rice yield to determine the optimum water level for crop-water level management and GHG emissions mitigation.

The work is well presented with some important outcomes.

Comments

Title: One of the important results of this study is GHG emissions mitigation. I am wondering why this is not captured in the title. I suggest the inclusion of GHG emissions mitigation in the title.

Line: Why the choice of these 6 locations? Are there any special consideration for selecting these locations?

Fig. 2. A legend to describe the parameters presented in the fig. will enhance the self-explanation of the figure. Also, different notations should be used for the average 129 monthly maximum and minimum temperatures.

Line 300: Since excess water above the optimum level could also affect the yield and GHG emission mitigation, provision for draining the excess water just as irrigation is equally important.

Line 336: Multiple drainage could lead to excess removal of the water beyond the desired optimum level. I am wondering how repeated drainage induced favourable conditions on the rice yield when it is postulates that water at optimum level improves the yield.

Author Response

Title: One of the important results of this study is GHG emissions mitigation. I am wondering why this is not captured in the title. I suggest the inclusion of GHG emissions mitigation in the title.

Following your suggestion, I modified the title to “Optimal water level management for GHG emissions mitigation under water-conserving irrigation in An Giang Province, Vietnam”

Line: Why the choice of these 6 locations? Are there any special consideration for selecting these locations?

We selected six locations in six districts in An Giang province where have been built the full-dikes for applying 3 crop seasons of rice farming. Even six sites located in the full-dike areas, but water regime and irrigation and drainage system may differ. We would like to see relations between water regimes with rice yield and CH4 emission, thus we selected locations with possible largest variations of water management.

Fig. 2. A legend to describe the parameters presented in the fig. will enhance the self-explanation of the figure. Also, different notations should be used for the average 129 monthly maximum and minimum temperatures.

Thank you for your suggestion, I modified figure legend as you suggested.

Line 300: Since excess water above the optimum level could also affect the yield and GHG emission mitigation, provision for draining the excess water just as irrigation is equally important.

Yes, that is sure and I added your comment in discussion part at Line 344.

Line 336: Multiple drainage could lead to excess removal of the water beyond the desired optimum level. I am wondering how repeated drainage induced favorable conditions on the rice yield when it is postulates that water at optimum level improves the yield.

The purpose of multiple drainage is suppling oxygen to plant root is important for plant growth. Oxygen directly or indirectly affects nitrification dominated by nitrifying bacteria, or chemical oxidation for the conversion of NH4+ to NO3− at the root surface. A mixed supply of NH4+ and NO3− to both upland and lowland rice cultivars resulted in significant increases in the dry weight and grain yield compared with application of either NH4+ or NO3− as the sole N source (Qian et al. 2004; Duan et al. 2007). Multiple drainage may be helpful to supply both NH4+ and NO3− Nitrogen sources for rice growth.  

We added some of the sentence about it in Chapter 4.2. Effect of multiple drainage on rice yield.

Reviewer 2 Report

It is necessary to describe the novelty of the work more in relation to the existing articles. There is a need to comment more on the benefits and weaknesses of existing publications. What was the motivation for this article? What are the previous articles lacking in this direction of research?

Comments:

1) Do not use abbreviations in the abstract.

2) Using a list of lumped references is not helpful to the readers. At least a short justification should be provided - individually. See "(Arai et al., 2021; Guo et al., 2017; Uno et al., 2021)."

3) Is there any logic to the selection of specific locations in Fig. 1?

4) Did the choice of the type of cultivated product in Tab. 1 have any logic?

5) To use the Pearson correlation coefficient, the condition for the data - a normal distribution of probabilities - must be met. If the condition of data normality is not met, it is appropriate to use Spearman's rank correlation coefficient.

6) In Fig. 3c is an extreme value (WS [-600, 8]) that can have a significant effect on the result. Try linear regression without this value. Similarly in Fig. 3a, Fig. 5a, and c.

7) The regression models in Fig. 3 to 5 show very small indices of determination. They describe only about 20% of the data variability (R^2). This means that there are other factors that influence the agricultural return. It would be appropriate to recommend other factors that would be good to measure during the experiment.

Author Response

1) Do not use abbreviations in the abstract.

Thank you for your suggestion, I removed abbreviations in the abstract as you suggested.

2) Using a list of lumped references is not helpful to the readers. At least a short justification should be provided - individually. See "(Arai et al., 2021; Guo et al., 2017; Uno et al., 2021)."

We modified lumped references with justification as follows:

These simplified water management systems also resulted in an in-crease in rice yield and reduction of GHG emissions in Mekong Delta [13], China [14], and Vietnam [15].

3) Is there any logic to the selection of specific locations in Fig. 1?

We selected six locations in six districts in An Giang province where have been built the full-dikes for applying 3 crop seasons of rice farming. Even six sites located in the full-dike areas, but water regime and irrigation and drainage system may differ. We would like to see relations between water regimes with rice yield and CH4 emission, thus we selected locations with possible largest variations of water management.

4) Did the choice of the type of cultivated product in Tab. 1 have any logic?

In this study, we follow practices of farmers; thus, we simply report cultivated products according to the farmers. Farmers apply rice varieties that may also differ in different areas. We would like to see relations between water regimes with rice yield. Thus we want to test whether the determined relation between water regimes with rice yield can be true for all rice varieties or not.

5) To use the Pearson correlation coefficient, the condition for the data - a normal distribution of probabilities - must be met. If the condition of data normality is not met, it is appropriate to use Spearman's rank correlation coefficient.

A normal distribution of probabilities was met in our data.

6) In Fig. 3c is an extreme value (WS [-600, 8]) that can have a significant effect on the result. Try linear regression without this value. Similarly in Fig. 3a, Fig. 5a, and c.

We modified Figures with your suggestion and modified figure was uploaded.

7) The regression models in Fig. 3 to 5 show very small indices of determination. They describe only about 20% of the data variability (R^2). This means that there are other factors that influence the agricultural return. It would be appropriate to recommend other factors that would be good to measure during the experiment.

Thank you for your suggestion, it is true R^2 was only 20%, however, P-value is less than 0.001. It is not experiment in the research center neither university, so we think that the water level management is a important factors to increase rice yield and GHG emissions from the paddy field.  

Reviewer 3 Report

In this paper, the authors analyzed a dataset obtained from a paddy water level management study carried out over a three-year period in An Giang province, Southern Vietnam. Overall, the authors have made a good attempt. However, the reviewer fails to understand the novelty of this work. The alternate wetting and drying (AWD) irrigation is not a new technology. Several researches have already performed in past studies. Besides, the reviewer has a big concern about high similarity of this paper. In iThenticate, the similarity is “39 %”. This is not acceptable as it is. The reviewer’s other comments are as follows:

1.      The authors should check and correct the author names.

2.      The authors should check and correct the e-mail address of the corresponding author. What’s “Abstract: A single”?

3.      The authors should not use acronym without explanation. All acronyms must be defined before use. e.g. GHG (Greenhouse Gas), etc.

4.      In the abstract part, the novelty and key idea of the proposed method should be described. The authors only described that “For the easy implementation of AWD to improve rice yield, we analyzed a dataset obtained from a paddy water level management study carried out over a three-year period with three cropping seasons at six locations (n = 82) in An Giang province, Southern Vietnam”. The novelty and key idea are not clear.

5.      The citation format of this paper does not follow the MDPI rules. The authors should follow the rules. The authors must quote the references according to the reference number. (You should start the quotation according to the reference number.)

6.      In the introduction part, the authors described that “Moreover, the way by which controlled optimized water level management at different N fertilizer application rates influences rice yield is yet to be addressed under water-conserving irrigation systems such as AWD and MD.” The reviewer disagrees with the authors’ opinion. For example, “S.M. Mofijul Islam, Yam Kanta Gaihre, Jatish Chandra Biswas, Md. Sarwar Jahan, Upendra Singh, Sanjoy Kumar Adhikary, M. Abdus Satter, M.A. Saleque, Different nitrogen rates and methods of application for dry season rice cultivation with alternate wetting and drying irrigation: Fate of nitrogen and grain yield, Agricultural Water Management, Volume 196, 2018, Pages 144-153, ISSN 0378-3774, https://doi.org/10.1016/j.agwat.2017.11.002.”, and so on. In-depth research survey is necessary”, and so on. You can find many related works so easily on the Internet.

7.      In the Introduction part, strong points of this proposed method should be further stated and organization of this whole paper is supposed to be provided in the end.

8.      The authors must justify the effectiveness of the proposed method by comparing with the conventional methods.

9.      The results of this research are not clear in Conclusions. The authors should show the new findings and scientific contribution of this work with concrete data. The authors described that “We observed significant relationships among the factors and found that specific water management practices could increase rice yield and reduce CH4 emissions.” This is trivial. Besides, authors described that “Control of the average water level during whole crop period has a potential for both yield increasing and reduce CH4 emissions. Furthermore, the results can be the basis for the dissemination of relatively simple optimal water level management practices in the Mekong Delta and regions with a similar climate and agronomic environment.” This interpretation is not supported by any demonstrations.

Author Response

Answers for Reviewer 3

In this paper, the authors analyzed a dataset obtained from a paddy water level management study carried out over a three-year period in An Giang province, Southern Vietnam. Overall, the authors have made a good attempt. However, the reviewer fails to understand the novelty of this work. The alternate wetting and drying (AWD) irrigation is not a new technology. Several researches have already performed in past studies. Besides, the reviewer has a big concern about high similarity of this paper. In iThenticate, the similarity is “39 %”. This is not acceptable as it is. The reviewer’s other comments are as follows:

Thank you for your comment and following your suggestion, we renewed the M&M parts to reduce similarity.

  1. The authors should check and correct the author names.

Thank you for your kind confirmation

  1. The authors should check and correct the e-mail address of the corresponding author. What’s “Abstract: A single”?

Thank you for your suggestion, we changed corresponding author and his e-mail address.

“Abstract: A single”was removed. 

  1. The authors should not use acronym without explanation. All acronyms must be defined before use. e.g. GHG (Greenhouse Gas), etc.

Thank you for your suggestion, we modified All acronyms.

  1. In the abstract part, the novelty and key idea of the proposed method should be described. The authors only described that “For the easy implementation of AWD to improve rice yield, we analyzed a dataset obtained from a paddy water level management study carried out over a three-year period with three cropping seasons at six locations (n = 82) in An Giang province, Southern Vietnam”. The novelty and key idea are not clear.

We modified as followed:

For the easy implementation of alternate wetting and drying for the farmers, we analyzed a dataset obtained from a farmer’s water management study carried out over a three-year period with three cropping seasons at six locations (n = 82) in An Giang province, Southern Vietnam.

  1. The citation format of this paper does not follow the MDPI rules. The authors should follow the rules. The authors must quote the references according to the reference number. (You should start the quotation according to the reference number.)

Citation format was modified to follow MDPI rules

  1. In the introduction part, the authors described that “Moreover, the way by which controlled optimized water level management at different N fertilizer application rates influences rice yield is yet to be addressed under water-conserving irrigation systems such as AWD and MD.” The reviewer disagrees with the authors’ opinion. For example, “S.M. Mofijul Islam, Yam Kanta Gaihre, Jatish Chandra Biswas, Md. Sarwar Jahan, Upendra Singh, Sanjoy Kumar Adhikary, M. Abdus Satter, M.A. Saleque, Different nitrogen rates and methods of application for dry season rice cultivation with alternate wetting and drying irrigation: Fate of nitrogen and grain yield, Agricultural Water Management, Volume 196, 2018, Pages 144-153, ISSN 0378-3774, https://doi.org/10.1016/j.agwat.2017.11.002.”, and so on. In-depth research survey is necessary”, and so on. You can find many related works so easily on the Internet.

Although ammonium (NH4+) is the primary form of available N in flooded fields, and rice prefers NH4+ over NO3− (Wang et al. 1993), physiological studies have shown that lowland rice is exceptionally efficient at acquiring NO3− through nitrification in the rhizosphere (Li et al. 2006; Duan et al. 2007). A mixed supply of NH4+ and NO3− to both upland and lowland rice cultivars resulted in significant increases in the dry weight and grain yield compared with application of either NH4+ or NO3− as the sole N source (Qian et al. 2004; Duan et al. 2007).

  1. In the Introduction part, strong points of this proposed method should be further stated and organization of this whole paper is supposed to be provided in the end.

Modified in introduction part.

  1. The authors must justify the effectiveness of the proposed method by comparing with the conventional methods.

Modified in M&M part.

  1. The results of this research are not clear in Conclusions. The authors should show the new findings and scientific contribution of this work with concrete data. The authors described that “We observed significant relationships among the factors and found that specific water management practices could increase rice yield and reduce CH4 emissions.” This is trivial. Besides, authors described that “Control of the average water level during whole crop period has a potential for both yield increasing and reduce CH4 emissions. Furthermore, the results can be the basis for the dissemination of relatively simple optimal water level management practices in the Mekong Delta and regions with a similar climate and agronomic environment.” This interpretation is not supported by any demonstrations.

Below-mentioned main conclusion was added:

The results offer insights that multiple drainage during whole crop period and nitrogen fertilization period have a potential for farmers implementation to contribute both yield increase and greenhouse gas emissions reduction from rice cultivation.

Reviewer 4 Report

Dear Authors,

The submitted manuscript titled „Optimal water level management to improve rice yield under water-conserving irrigation in An Giang Province, Vietnam” contains valuable results. The work is adressed to very important topic of improvement of rice yield and therefore I am convinced, that it might interst an international audience.  However, I have found some imperfections- which in my opinion- should be improved or clarified before an eventual publication. Please, find them below:

1.       The Abstract section- please add the main conclusion/s.

2.       Introduction –the current state of knowledge on relations of water level and rice yield should be supplemented.

3.       Material and methods

·         In my opinion the Figure showing the experimental design would be very useful in understanding the applied methods.

·         How many plots was investigated?

·         Are there any repetitions?

·         Subchapter 2.3 How many measurement was made?

4.       Figures 3-6 are illegible.

5.       In my opinion Discussion chapter might be enriched in references. Please, look into below listed literature sources. Perhaps, some of them will be useful in a manuscript corrections.

·         Cabangon, R.J., Tuong, T.P., Castillo, E.G. et al. 2004. Effect of irrigation method and N-fertilizer management on rice yield, water productivity and nutrient-use efficiencies in typical lowland rice conditions in China. Paddy Water Environ 2, 195–206.

·         X.L. Cai, B.R. Sharma. 2010. Integrating remote sensing, census and weather data for an assessment of rice yield, water consumption and water productivity in the Indo-Gangetic river basin. Agricultural Water Management, 97,2, 309-316.

·         B.A.M. Bouman, Liping Feng, T.P. Tuong, Guoan Lu, Huaqi Wang, Yuehua Feng 2007. Exploring options to grow rice using less water in northern China using a modelling Approach: II. Quantifying yield, water balance components, and water productivity, Agricultural Water Management, 88, 1–3, 23-33.

·         Xue, C., Yang, X., Bouman, B.A.M. et al. 2008. Optimizing yield, water requirements, and water productivity of aerobic rice for the North China Plain. Irrig Sci 26, 459–474.

·         Kaiming Liang, Xuhua Zhong, Nongrong Huang, Rubenito M. Lampayan, Junfeng Pan, Ka Tian, Yanzhuo Liu, 2016. Grain yield, water productivity and CH4 emission of irrigated rice in response to water management in south China, Agricultural Water Management, 163, 319-331.

Author Response

  1. The Abstract section- please add the main conclusion/s.

Below-mentioned main conclusion was added:

The results offer insights that multiple drainage during whole crop period and nitrogen fertilization period have a potential for both yield increase and greenhouse gas emissions reduction from rice cultivation.

  1. Introduction –the current state of knowledge on relations of water level and rice yield should be supplemented.

Thank you for your suggestion, with your suggestion, we added reference in discussion part.

  1. Material and methods
  • In my opinion the Figure showing the experimental design would be very useful in understanding the applied methods.
  • How many plots was investigated?

Are there any repetitions?

Subchapter 2.3 How many measurement was made?

In total 82 fields with three replications (plots)

Measurement was conducted three times for each field.

We modified M&M chapter including with your suggestion

  1. Figures 3-6 are illegible.

Figure was modified to be legible.

  1. In my opinion Discussion chapter might be enriched in references. Please, look into below listed literature sources. Perhaps, some of them will be useful in a manuscript corrections.

Thank you for your suggestion, with your suggestion, we added reference in discussion part.

Round 2

Reviewer 2 Report

I consider the revision of the text to be sufficient.

Author Response

We thank that the reviewer considers the revision of the text to be sufficient. 

Reviewer 3 Report

In this paper, the authors analyzed a dataset obtained from a paddy water level management study carried out over a three-year period in An Giang province, Southern Vietnam. Thank you for submitting the revised version of the paper ID: sensors-1964876. In the revised version, most of the reviewer’s requests were met by the authors. However, some of them are not improved yet. Especially, the reviewer has a big concern about the high similarity of this manuscript.

1.       The similarity of this paper is still high. In iThenticate, the similarity is “34 %”. The authors should improve the presentation.

2.       There is no response for the reviewer’s comment #7: “In the Introduction part, strong points of this proposed method should be further stated and organization of this whole paper is supposed to be provided in the end.

Author Response

We thank that the reviewer considers the manuscript received sufficient improvement.

  1. The similarity of this paper is still high. In iThenticate, the similarity is “34 %”. The authors should improve the presentation.

Answer: According to the comment on the high similarity of this paper, we used Turnitin to scan to detect the similarity points, and then we revised it. After revising, we checked the similarity, which is below 18%. We would appreciate if the reviewers would accept the similarity at this level.

  1. There is no response for the reviewer’s comment #7: “In the Introduction part, strong points of this proposed method should be further stated and organization of this whole paper is supposed to be provided in the end.”

Following your suggestion, we modified several sentense from Line 134 to 151.

Reviewer 4 Report

Dear Authors,

In my opinion the manuscript received sufficient improvement, thus I do not have any further remarks.

Author Response

We thank that the reviewer considers the manuscript received sufficient improvement.